# Possible nitrogen fertilization of the early Earth Ocean by microbial continental ecosystems

Christophe Thomazo (ID) [1], Estelle Couradeau (ID) [1,2] & Ferran Garcia-Pichel[2]

While significant efforts have been invested in reconstructing the early evolution of the Earth's atmosphere–ocean–biosphere biogeochemical nitrogen cycle, the potential role of an early continental contribution by a terrestrial, microbial phototrophic biosphere has been largely overlooked. By transposing to the Archean nitrogen fluxes of modern topsoil communities known as biological soil crusts (terrestrial analogs of microbial mats), whose ancestors might have existed as far back as 3.2 Ga ago, we show that they could have impacted the evolution of the nitrogen cycle early on. We calculate that the net output of inorganic nitrogen reaching the Precambrian hydrogeological system could have been of the same order of magnitude as that of modern continents for a range of inhabited area as small as a few percent of that of present day continents. This contradicts the assumption that before the Great Oxidation Event, marine and continental biogeochemical nitrogen cycles were disconnected.

[1] Biogéosciences UMR6282, CNRS, Univ. Bourgogne Franche-Comté, 21000 Dijon, France. [2] Center for Fundamental and Applied Microbiomics and School of Life Sciences, Arizona State University, 85282 Tempe, AZ, USA. Correspondence and requests for materials should be addressed to C.T. (email: Christophe.thomazo@u-bourgogne.fr)

In the last decade significant progress has been made in reconstructing the evolution of the nitrogen (N) biogeochemical cycle through Earth's history, with an increasing focus on the Archean period, from 4.0 to 2.5 Ga[1–4]. A general time frame (Fig. 1) emerged from these recent contributions that proposes the following sequence of events: from 3.6 to 3.2 Ga during the Paleoarchean, microbial diazotrophy evolved and supported the nitrogen demand of the earliest ecosystems[3,5], between 2.7 and 2.6 Ga an oxic pathway of the nitrogen cycle emerged even before full atmospheric oxygenation at 2.45 Ga during the Great Oxidation Event[6–8] (GOE), at 2.3 Ga nitrate became more widely available due to ocean oxygenation[9], from 1.9 Ga onwards, during the mid-Proterozoic, aerobic and anaerobic nitrogen cycling were spatially separated between surface and deep waters[1,10–12], and eventually, around 750 Ma (but possibly back to 1 Ga) during the Neoproterozoic, there arose a modern nitrogen cycle featuring widespread aerobic activity[13,14]. This sequential evolution of the N cycle involving changes in the balance of N source and sink fluxes has been tentatively linked to important atmospheric pressure fluctuations observed during the Precambrian[15]. The main drivers of these changes are biological evolution[16] along with redox variations in Earth's surface environments, primarily due to the oxygenation of the atmosphere–ocean system[1,2].

This scenario is largely based on the interpretation of secular variation of the nitrogen isotopes from the sedimentary rock record[1,2,9]. Physical, chemical, and biological processes differentially incorporate the two stable N isotopes ($^{15}N$ and $^{14}N$), leading to measurable differences in the $^{15}N/^{14}N$ ratio of sedimentary nitrogen. Nitrogen isotopes can provide a record of specific biosignatures and are sensitive to environmental redox changes[17], making them a good proxy to track the evolution of N reservoirs and fluxes throughout Earth history. However, their use is challenging due to a number of technical pitfalls, including the fact that nitrogen is found and preserved in measurable quantities in relatively few phases, primarily organic matter, phyllosilicates, feldspars, magnetite, and fluid inclusions (a compilation of Precambrian $\delta^{15}N$ data is in ref. [1]). Another limitation arises from the scarcity of the Archean sedimentological record (compared to that of the Phanerozoic) and the even poorer archive of continental paleosols. Altogether, this led to the use of geochemical box models to reconstruct the Precambrian balance of N source and sink fluxes without taking into account the possible role of an evolving continental biosphere reservoir[15,18].

During the Archean, the lack of an ozone layer resulted in higher short-wavelength irradiance than today despite the fact that the sun was 30% dimmer[19]. Because of this peculiar environmental condition Berkner & Marshall[20] first postulated that the

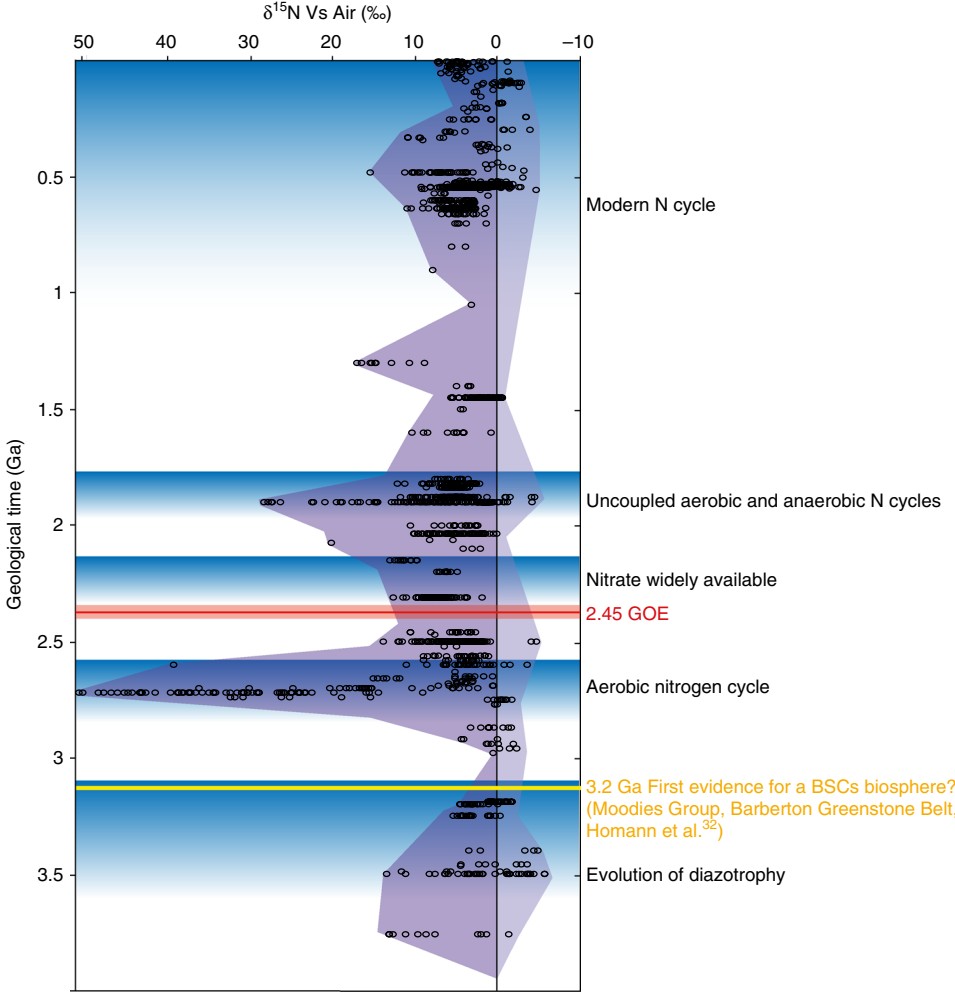

**Fig. 1** Secular variation of the sedimentary $\delta^{15}N$ record through geological time. The stepwise evolution of the biogeochemical nitrogen cycling is shown as blue shaded areas. The Great Oxidation Event (GOE) and first evidence of terrestrial biological soil crusts (BSCs) are shown in red and yellow, respectively ($\delta^{15}N$ data compilation after refs. [1,9,74])

colonization of the landmasses was not possible before the formation of an ozone shield. However, recent findings demonstrate that in the primitive anoxic atmosphere sulfur vapor composed of sulfur molecules and hydrocarbon smog may have strongly attenuated ultraviolet radiation[21]. Also, a high concentration of ferrous ion (Fe II) may have been present in anoxic waters to significantly screen UV radiation[22]. Moreover, Cockell & Raven[23] show that under a worst-case UV flux (no environmental UV screen) on the Archean Earth, the landmasses could have been colonized by early photosynthetizers. Therefore, despite a "faint young sun" and the absence of a UV-protective ozone layer in the Archean[24,25] a terrestrial phototrophic biosphere composed of systems similar to modern biological soil crusts (BSCs) may have existed early, before the GOE[26], and could have colonized the exposed land surfaces[27].

Modern BSCs are arguably the most extensive biofilm on the planet covering up to 12% of Earth's continental area[28], and while they are composed of a wide diversity of microorganisms they are primarily built by cyanobacteria performing oxygenic photosynthesis[29]. With fresh volcanic material to act as fertile soil, in the absence of plant root systems and grazers, one could entertain the idea that BSCs should have had the potential to rapidly develop and thrive during Archean time due to lack of competition, even beyond their restricted modern habitat, which is circumscribed to areas devoid of extensive plant cover. The oldest direct evidence for fossil BSCs comes from the 1.2 Ga

mid-Proterozoic Apache Supergroup in the Dripping Springs Formation of Arizona, including sedimentological evidence for microbially induced sedimentary structures (MISS) and cyanobacteria-like organic microfossils[30]. But, indirect evidence of BSCs developing on Archean coastal plain paleosols date them back to 3.0 Ga[31]. Tufted microbial mats inhabiting coastal habitats have also been described within the Moodies Group (South Africa) at 3.2 Ga[32]. Chemical pieces of evidence based on element mobility patterns in several paleosols also suggest the presence of an ancient terrestrial biosphere where organic ligands chelated metals during weathering[33,34]. Modern BSC do indeed produce such chelators[35], and are known to mobilize metals[36]. The evidence timeline for BSC development is consistent with results of molecular clock estimates, that place the colonization of land at the latest between 3.05 and 2.78 Ga based on shared properties of pigment synthesis and resistance to desiccation exhibited by typically terrestrial extant bacterial phyla[37,38]. The impact of a pre-GOE BSCs colonization of the continents has already been explored in the framework of atmospheric oxygen concentration evolution. Pre-GOE BSC-driven photosynthetic oxygen flux to the atmosphere and ensuing oxidative weathering efficiency was proposed as a scenario to explain the existence of transient episodes of mild environmental oxygenation (so called "whiffs of oxygen") and oxidative continental weathering[26,39,40]. This scenario echoes theoretical predictions of a long lasting, mass-independent sulfur isotopic

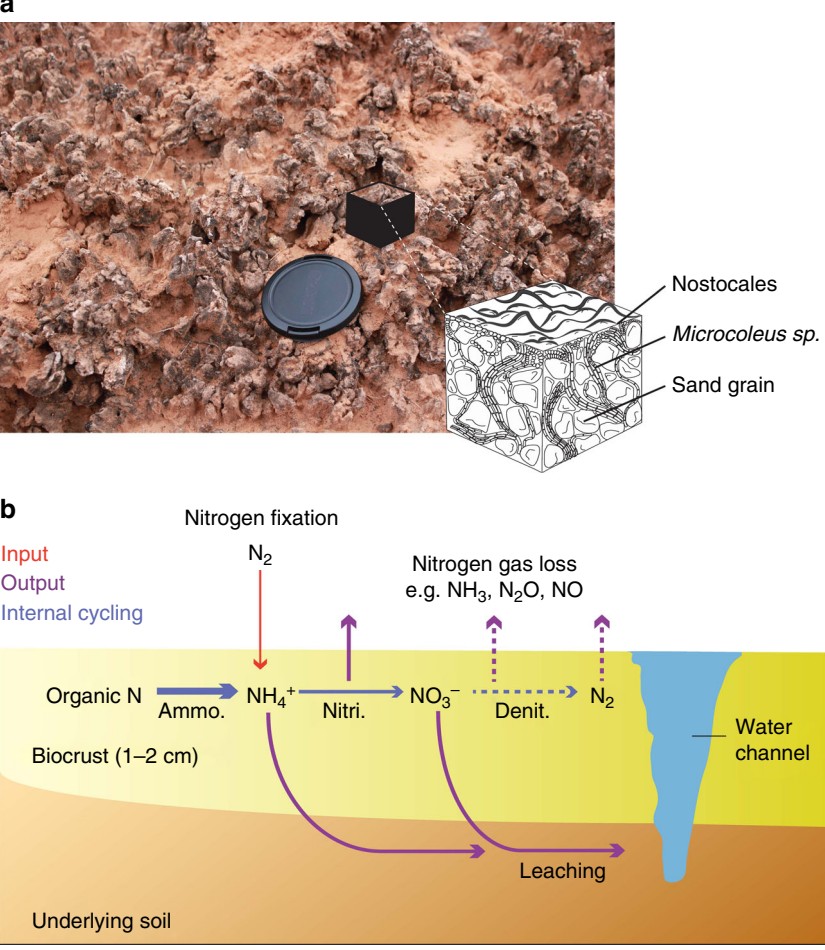

**Fig. 2** Biological soil crusts and associated nitrogen cycling. Modern biological soil crusts (**a**), and a scheme of its internal structure showing close association between filamentous cyanobacteria and mineral grains. Schematic illustration of nitrogen cycling transformations in modern biological soil crusts (**b**), reprinted/adapted by permission from ref. [44]

signal due to long-term sedimentary recycling of rare sulfur isotope anomalies[41].

Here, we address at a theoretical level the impact of early colonization of Archean continental landmasses by BSCs in terms of N species source fluxes to the ocean. We show that the colonization of a minor fraction of Archean landmasses by BSCs should have sufficed to attain the modern equivalent land-to-ocean N export flux of ammonium and nitrate. We suggest that the ocean could have acted as a net sink for ammonium and nitrate during the Archean before oceanic nitrate concentration reached a steady state at around 2.3 Ga.

## Results

**Nitrogen export capacity of modern BSCs**. Modern BSCs play a significant role in the N cycle within arid and semiarid ecosystems, as they contribute to major N inputs via biological fixation[42], harbor intense internal N transformation processes, and direct N losses via dissolved, gaseous, and erosional processes (Fig. 2). Since denitrification rates within BSCs are extremely low and because of limited respiration and ammonia oxidation, BSCs function as net exporters of ammonium, nitrate, and organic N to the soils they cover, in approximately stoichiometrically equal proportions[43]. Importantly, N cycling by BSCs likely regulates short-term soil N availability, as well as long-term N accumulation and maintenance of soil fertility[43]. Whereas multiple studies have examined a single compartment of the modern BSCs N budgets, the lack of studies that simultaneously address N inputs, losses, and soil pools, preclude the construction of definitive N balances (see a review of estimate fluxes of these processes in ref. [44]). Nonetheless, Walvoord et al.[45] reported on the existence of a large reservoir of nitrate beneath desert soils, that clearly indicates long-term net export of N from topsoil to arid land subsoils. Some studies give direct estimates of this export flux of N and leachates from BSCs colonizing beach dunes at the Indiana Dunes National Lakeshore along southern Lake Michigan in Indiana, USA[46] and at the Colorado Plateau, USA[43]. Both studies simulate rain events and measured resulting net N outputs from crusts to subsurface soil. The N outputs from crusts of the Colorado Plateau ranged from 0.98 to 5.02 g $NH_4^+$ N m$^{-2}$ yr$^{-1}$ and 1.10 to 3.8 g $NO_3^-$ N m$^{-2}$ yr$^{-1}$, commensurate with rates of N-fixation at the study site, although this was a measurement carried out at a maximal depth of only 20 cm below the crusts. At Indiana Dunes National Lakeshore, the net N outputs from crusts to subsurface soil were measured after a long-term greenhouse experiment in which intact soil cores composed of a population of mixed cyanobacteria, algae, and bryophytes forming BSCs were subjected to artificial rainfall over a full growth season[46]. Fluxes there ranged from 0.01 to 0.19 g $NH_4^+$ N m$^{-2}$ yr$^{-1}$ and 0.01 to 0.61 g $NO_3^-$ N m$^{-2}$ yr$^{-1}$. Accordingly, in this system the total inorganic N inputs to Lake Michigan sand dune ecosystem from BSCs ranged from 0.02 g N m$^{-2}$ yr$^{-1}$ to 0.8 g N m$^{-2}$ yr$^{-1}$. These two studies give congruent estimates of N export (slightly larger for the Colorado Plateau BSCs) and indicate the presence of net production zones of nitrate in all crusts. However, because incubations used by Johnson et al.[43] were performed on short timescale (20 days) the authors acknowledge that their outputs fluxes might be slightly overestimated due to non steady-state situation with respect to production, consumption and diffusion of N species. In the experiment designed by Thiet et al.[46], N was measured after it had leached beyond a depth of 7 cm and for some 75 days. For these reasons the measurements of N production and leachates of BSCs given by Thiet et al.[46] probably represent a more realistic appraisal. Estimates of N outputs from BSCs to subsurface soil can also be evaluated indirectly using the rule of thumb that 19–28% of the nitrogen fixed by soil crusts ends up in the underlying topsoil[47,48]. Biological soil crusts in cold deserts are estimated to fix 1–10 g N m$^{-2}$ yr$^{-1}$ [49,50], while estimates for the Sonoran desert and Australia range from 0.7–1.8 g N m$^{-2}$ yr$^{-1}$ and 0.13 g N m$^{-2}$ yr$^{-1}$, respectively[51,52]. The full range of indirect estimates is indeed quite large, ranging from 0.07 to 10 g N m$^{-2}$ yr$^{-1}$ [53–60]. Quantification of N fixation by biocrusts in a global survey yielded a geometric mean value of 0.76 g N m$^{-2}$ yr$^{-1}$ [42]. Using these range of estimates and a minimum export rate of 19%, we could scale indirect estimates of N crust output to subsurface soil from 0.013 to 1.9 g N m$^{-2}$ yr$^{-1}$ with a mean of 0.144 g N m$^{-2}$ yr$^{-1}$ (Table 1).

**Parameters driving the fate of nitrogen: past and present**. Additionally and importantly for our tentative deep-time transposition, Thiet et al.[46] demonstrated that the proportional cover of cyanobacteria and mosses did not significantly affect these fluxes. An increase in light intensity directly increased the N throughput, while its decrease when combined with an increase in rainfall intensity indirectly increased the N output fluxes by maximizing the leachate volume. Moreover, while it had been postulated that the export range might also depend on the BSCs ecosystem structure and especially the proportion of N-fixing organisms and their N-fixation rates[61] compared with the

**Table 1 Compilation N outputs estimates from biocrusts to subsurface soil from various location worldwide and the respective percentage of Archean land crust cover needed to reach the modern N land-to-ocean export flux**

|  | Location | Net N outputs from crusts to subsurface soil (m$^{-2}$ yr$^{-1}$) | Percentage of Archean land coverage needed to reach the modern N export flux |
|---|---|---|---|
| Johnson et al.[43] | Colorado Plateau | 2.080–8.820[a] | 1.13–0.27% |
| Thiet et al.[46] | Lake Michigan | 0.020–0.800[a] | 117.8–2.94% |
| Rychert & Skujins[49]; West & Skujins[50] | Cold desert | 0.19–1.9 | 12.4–1.24% |
| Rychert et al.[51] | Australia | 0.025 | 94.2% |
| Evans & Johansen[52] | Sonoran Desert | 0.133–0.342 | 17.71–6.88% |
| Evans & Lange[53]; Belnap[54, 55]; Russow et al.[56]; Stewart et al.[57–59]; Caputa et al.[60] | Global | 0.0133–1.9 | 177–1.24% |
| Elbert et al.[42] | Global | 0.144[b] | 16.35% |

The modern export flux is set to 0.15 Tmol yr$^{-1}$ based on N/C data from shales and continental organics[15]. Detailed calculations are described in the methods section and available in Supplementary Table 1
[a]Direct estimate of N export fluxes from modern biocrusts, all other values of N outputs from crusts to subsurface soil are extrapolated from the nitrogen import fluxes
[b]Geometric mean value given by Elbert et al.[42]

proportion and activity of denitrifiers, recent direct evidence shows that these factors are unimportant[62]. Using modern desert BSCs obtained across biogeographic regions in the Southwestern United States from the Colorado Plateau and the Mojave, Sonoran and Chihuahuan Deserts, Strauss et al.[62] showed that the measured rates of major biogeochemical N-transformations (N-fixation, aerobic ammonia oxidation, anammox, and denitrification) despite the demonstrable differences in microbial community composition and soil material displayed a remarkably consistent pattern of internal N cycling. Dinitrogen fixation and aerobic ammonia oxidation were prominent transformations at all sites, while anaerobic ammonia oxidation (anammox) rates were below the detection limit in all cases, and heterotrophic denitrification was also of little consequence for the flow of N, with rates at least an order of magnitude smaller than those of N-fixation. These modern examples of BSCs from sandy soils are good analogs of the environment available for the development of BSCs on Archean continents given the aggressive weathering regime postulated for the early Earth[63] when sedimentation was largely dominated by siliciclastic inputs[64], and with aeolian sand systems reported as early as 3.2 Ga (Lower Moodies Group of the Swaziland Supergroup in South Africa[32]). Environmental parameters may therefore primarily control the export rate in BSCs over a wide range of microbial community composition and soil material, and the main N escape routes are likely to depend more on hydrology than on biology. While the fate of these exports remains to be assessed, it is likely that soil bacterial populations, depending on available oxidants, will rapidly use organic N and ammonium. But, nitrate exports are probably of longer range, because the molecule is environmentally mobile and biologically less desirable than ammonium.

**Quantitative modeling**. On the geological timescale of interest here, a hypothetical contribution of BSCs in transferring atmospheric fixed $N_2$ to the ocean in the form of organic nitrogen, ammonium and nitrate (Fig. 2) may have had consequences for the evolution of the global N cycle, which would have eventually been recorded in the $\delta^{15}N$ secular variation. This prompted us to evaluate the pertinence of this hypothesis quantitatively by modeling the fluxes of N that may have been exported to the underlying soil and hydrogeological system by BSCs using the fluxes obtained in modern studies. For the sake of clarity, we express those as a function of a percent of colonized modern continental area or Archean equivalent landmasses (assuming a total emerged surface of about 90 Mkm² at around 2.45 Ga, i.e., 60% of the present day value[65]). We hypothesized that N outputs from BSCs to subsurface soil and hydrogeological system were delivered to the oceanic basin after weathering, without further remobilization. This first-order hypothesis probably tends to overestimate N transport to the ocean but is consistent with Archean BSCs developing in coastal habitats close to oceanic basins[31,32] and with a reduced soil biological demand. The results are presented in Table 1 and Fig. 3 as the N transport of inorganic N for a variety of net N throughput scenarios based on the direct and indirect results from modern BSCs studies. We calculate that the total inorganic N transport is likely to have been quite substantial, reaching the proposed value for the Phanerozoic land-to-ocean N flux (around $0.15 \pm 0.03$ Tmol yr$^{-1}$ based on N/C data from shales and continental organics[15]) when Precambrian areas colonized by BSCs attained anywhere from <1% to more than 113% of the modern continental area if one includes the full range of modern net N outputs (from 0.0133 to 8.820 g N m$^{-2}$ yr$^{-1}$, Supplementary Table 1). Table 1 summarizes the literature estimates that were used for our calculations. It shows that published estimates vary by more

than two orders of magnitude. Based on the available data, however, all but three reported values support the idea that the colonization of a minor fraction of Archean landmasses by BSCs (<18%) would suffice to attain the modern equivalent N export flux. More precisely, the geometric mean value computed by Elbert et al., which is the most robust global estimate available to date, suggests that 16% of land colonization by biocrust would be enough to recapitulate the modern N export. Overall, in both modern and Archean configurations, a high to moderate range of net N outputs from BSCs to subsurface soil of 0.8 to 0.1 g N m$^{-2}$ yr$^{-1}$ (with a mean indirect estimate of 0.144 g N m$^{-2}$ yr$^{-1}$ [42]) implies a total inorganic N transport comparable to Phanerozoic land-to-ocean N fluxes at a rather small continental cover ($\approx$3 to 25% based on Thiet et al.[46] direct estimates and 16% based on Elbert et al.[42] indirect estimate) (Fig. 3). Given that current cover by crusts, even when their current potential habitat is restricted to lands with sparse higher plant vegetation, is some 12% of the continents[28], those levels of Precambrian cover seem plausible.

## Discussion

These results are based on the assumption that modern analogs are fair proxies to ancient systems and thus may suffer from some pitfalls, especially when considering the high uncertainty in the exact nature of the Archean paleoenvironment, including aspects such as the range of air pressure[15,66,67] or temperature[68]. Based on 2.73 Ga fossil raindrop imprint[67], Archean Earth air pressure estimates range from 0.52 to 1.1 bar; nitrogen and argon isotopic studies suggest a $pN_2$ upper limit of 0.5–1.2 bar at around 3.0 to 3.5 Ga[66]; pressure calculated from subaerial lava flows a lower sea-level air pressure at 2.74 Ga at $0.23 \pm 0.23$ bar[15]. However, even with significant atmospheric $pN_2$ swings on the early Earth, it is unlikely that nitrogen could become a limiting factor to nitrogen fixation-based microbial primary productivity[3,5]. Future works are nevertheless needed to better-characterized variations in the kinetic rate of N-fixation with changing pressure temperature.

Moreover, the fate of Archean N outputs in the form of nitrate may have also differed from those in the present day surface environment due to different redox conditions. Indeed, it is well established that before 2.4 Ga the early Earth was anoxic with atmospheric $pO_2$ below $10^{-5}$-fold the present atmospheric level[69]. The existence of oxygen oases in restricted ecosystems (e.g., lake, water bound) is nonetheless often argued to account for the occurrence of redox cycling of elements (including nitrogen but also, sulfur, carbon, and metallic elements[26]). In any case, while we could speculate that the Archean low $O_2$ concentration might have favored the removal of nitrate from the hydrological system to the atmosphere by denitrification (early on, close to the production site, or later and more distally in the ocean), even today[43,62] denitrification is not highly relevant to modern BSCs. Therefore, we suggest here that accumulation and transfer of nitrate and ammonium in the Archean continental system is a plausible hypothesis worthy of attention and further investigation.

Ultimately, the proposed transfer of N from the continent to the ocean might not have had long-term consequences for the atmospheric $N_2$ reservoir size because BSCs and the biosphere in general is a minute, transient reservoir, and the only forms of N that can accumulate over geologic timescales are $N_2$ in the atmosphere and reduced N in rocks. It has been shown that carbon residence time in BSCs is very short in geological scales[42]. However, the flux of fixed N from BSCs would be significantly larger than that from abiotic sources[70] and it could have helped overcome the limitation of early biosphere to fixed N using

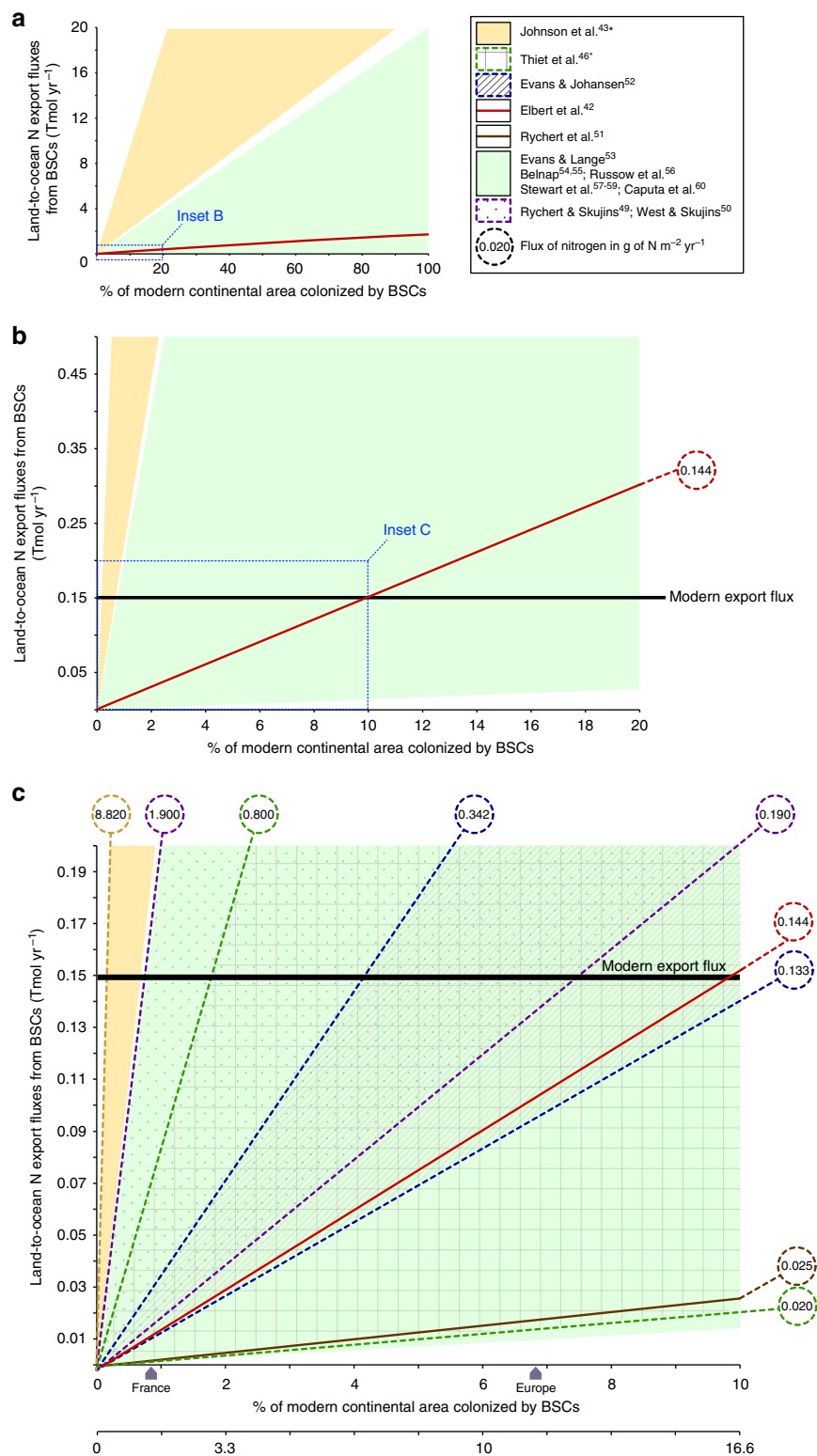

**Fig. 3** Modeling results of the land-to-ocean N export fluxes from biological soil crusts as a function of the percentage of modern continental area colonized, for a variety of net N transport fluxes. **a** Modeled export fluxes (Tmol yr$^{-1}$) from biological soil crusts (BSCs) as a function of the percentage of modern continental area colonized (from 0 to 100%) for a variety of net N transport fluxes to underlying soil within the range of estimates for modern BSCs. **b** Close in from 0 to 20% continental cover. **c** Close in from 0 to 10% continental cover. The upper X-axis shows the percentage of modern continental area, the size of France and Europe are included as an intuitive benchmark. The lower X-axis shows the corresponding percentage of Archean continental area. The Phanerozoic N flux from land into the ocean (0.15 ± 0.03 Tmol yr$^{-1}$) is denoted by the horizontal black bold line. Asterisks denote data from direct measurements of N export fluxes, all other values are indirectly obtained from nitrogen import fluxes (see Methods)

diazotrophy[71] and fuel primary productivity with newly available nitrogen species.

The delivery of nitrate and ammonium to the hydrogeological reservoir would also have provided a new evolutionary driver and an opportunity to trigger diversification of prokaryotes and eukaryotic phytoplankton able to use fixed nitrogen as a source of nutrient before nitrate concentrations stabilized in the ocean at around 2.3 Ga. As a corollary, BSCs may have enabled the return of nitrogen to the atmosphere via nitrate released to the ocean well before the GOE. Given the potential importance of this transfer compared to estimates of the Phanerozoic oxidative weathering N flux, an early Archean development of BSCs should be taken into account when modeling Archean surface atmospheric pressure[15] and $pN_2$ secular variation[2,18].

Finally, a complex redox inorganic nitrogen cycle involving ammonium, nitrite, and nitrate may have arisen in Archean paleosols. Here, we calculate that this niche potentially delivered significant quantities of ammonium and nitrate to the ocean before the GOE. Secular variation of sedimentary $\delta^{15}N$ in fact is consistent with these hypotheses, as lines of evidence exist for the presence of aerobic, nitrate-based nitrogen cycling in Meso to Neo-archean sediments from 2.7 to 2.5 Ga[6–8,72] (Fig. 1). There is also evidence for late mesoarchean methanotrophs using nitrate in ephemeral ponds and shallow water environments[33,73]. While better quantification needs to be achieved and the potential feedback of such changes on the early Earth system is yet to be thoroughly evaluated, our results speak for a renewed relevance of future research on the Archean record of BSCs, bringing together the community working on the biogeochemistry of modern BSCs with deep-time biogeochemists.

## Methods

**Flux calculations**. The N export fluxes from BSCs (Fs; Tmol yr$^{-1}$) was calculated using the full range of direct and indirect (compilation of nitrogen input times 19% of export) estimates of N transport fluxes ($F$) to underlying soil (between 0.0133 and 8.820 g N m$^{-2}$ yr$^{-1}$, see Table 1) and integrated over the surface of colonized continental (S; m$^2$) using the following equation: $FS = \frac{F \times 10^{-12} \times S}{M}$, where $M$ is the molar mass of nitrogen species. Calculated surface was subsequently scaled as percentages of either the modern or Archean landmasses.

**Data availability**. The authors declare that the data supporting the findings of this study are available within the paper and its Supplementary Information files.

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

## Acknowledgements

This project was funded by the European Union's Seventh Framework Programme for research, technological development and demonstration under grant agreement no 328530. We thank Arnaud Brayard and the members of the GPI for fruitful discussions.

## Author contributions

C.T., E.C. and F.G.-P. designed the project, participated to the data analysis, contributed to the biogeochemical N cycle interpretations, and wrote the manuscript.

## Additional information

**Competing interests:** The authors declare no competing interests.

