## [Peer Review File · Nature Communications]

Editorial Note: Parts of this peer review file have been redacted as indicated to remove third-party material where no permission to publish could be obtained.

Reviewers' comments:

Reviewer #1 (Remarks to the Author):

Review of: Thomazo et al. "Possible nitrogen fertilization of the early Earth ocean by continental biomass"

The authors develop the idea that bacterial soil crusts (BSC) could have led to a significant flux of fixed nitrogen (ammonium, nitrate) to the Precambrian ocean. The conclusion is supported by calculations based on extrapolations from modern BSC-related nitrogen fluxes.

The idea is novel and provocative. The nitrogen cycle has gained increasing interest over the last decade, which would make this contribution very timely. I would expect the paper to be highly cited.

I only have a few comments that should be addressed:

- One weakness of the calculations is that they are based on data from only one modern study of BSC (Thiet et al. 2005). However, that paper actually cites several other studies that report similar fluxes (see data on the Colorado Plateau, the Sonoran desert and Australia quoted on page 244). It would strengthen this new study if those values were quoted as well to support the decision of using a N flux of 0.02-0.8 as a reference range. Regarding this range, it would be very useful to know what was the main driver for the low and high end, so one can evaluate which values are more common.

- Another concern is whether Precambrian BSCs would indeed have produced as much nitrogen as modern ones. Most of the fixed nitrogen is probably released into soil when the bacterial biofilm is decomposed and cellular organic nitrogen gets released. To what degree is that process dependent on atmospheric O₂ and macrofauna? This may be difficult to assess, but it would be worth adding a comment to the main text about how nitrogen is released from BSCs into the environment.

- L. 38-39: Change to "from 1.9 Ga onwards...". The study by Stüeken (2013, ref. 10) is about the 1.45 Ga Belt Supergroup. You could also add a citation to Koehler et al. (2017, GCA), who reported similar spatial trends in two other basins at 1.5 Ga.

Koehler, M.C., Stüeken, E.E., Kipp, M.A., Buick, R. and Knoll, A.H., 2017. Spatial and temporal trends in Precambrian nitrogen cycling: a Mesoproterozoic offshore nitrate minimum. *Geochimica et Cosmochimica Acta*, 198, pp.315-337.

- L. 82: Another relevant reference would be Blank & Sanchez-Barcaldo (2010) who proposed that cyanobacteria first evolved in fresh water.

Blank, C.E. and Sanchez-Barcaldo, P., 2010. Timing of morphological and ecological innovations in the cyanobacteria—a key to understanding the rise in atmospheric oxygen. *Geobiology*, 8(1), pp.1-23.

- L. 89: If you want, you could add a citation to Stüeken et al. (2012, *Nature Geoscience*), which also concluded oxidative life on land.

Stüeken, E.E., Catling, D.C. and Buick, R., 2012. Contributions to late Archaean sulphur cycling by life on land. *Nature Geoscience*, 5(10), p.722.

- L. 141: The term 'oxidative weathering' is misleading. The BSC flux is not a weathering flux, because it does not affect the flux of nitrogen out of rocks. Better change to "... the Phanerozoic N flux from land into the ocean...". Also change 'canonical' to 'proposed'.

- L. 143, 146: You probably don't need to call it weight percent (wt. %), if it refers to a surface

area?

- L. 143 and Fig. 3: The 80% value is not shown on the x-axis in Fig. 3. At first, I didn't know where it came from. It would be good if you could show a small inset in Fig. 3 where the x-axis goes to 100%, such that one can see the intercept with the 0.02 gN/m²/yr flux.

- L. 167: Change to "...Phanerozoic N fluxes from land into the ocean, ...". As above, the term oxidative weathering is not quite correct here.

- Fig. 1 seems to be missing the new data from Zerkle et al. (2017 Nature) at 2.3 Ga. Since those are discussed in the text, they would be worth adding. For the Phanerozoic, I recommend adding the database (available online) from Algeo et al. (2014 Biogeosciences)

Algeo, T.J., Meyers, P.A., Robinson, R.S., Rowe, H. and Jiang, G.Q., 2014. Icehouse–greenhouse variations in marine denitrification. *Biogeosciences*, 11(4), pp.1273-1295.

Best wishes,
Eva Stüeken

Reviewer #2 (Remarks to the Author):

The paper claims to provide an alternative source of nitrogen to Archean oceans before the Great Oxidation Event based on a simple calculation using nitrogen export from one study of modern biological soil crusts, as a function of continental land coverage of BSCs.

The concept of terrestrial biological soil crusts as an Archean ecosystem is certainly worthy of further study, and the paper does a good job of citing the important nitrogen isotopic literature in this field of study.

However, this paper has a number of major flaws that make it unsuitable for publication.

The most major flaw is that the paper's nitrogen export values, which form the basis of the calculations, are from a SINGLE study of BSCs from beach sand dunes on Lake Michigan (Thiet et al., 2005), despite the author's citation of a book chapter (Barger et al., 2016) with section 14.3 ("Nitrogen Release to the Surrounding Substrate") citing numerous (~20) papers on the subject.

Other major problems with the study approach relate to the few variables examined in the calculations. The only factor that was included in the model was fraction of land area for Archean continents. There is no mention of the variability of nitrogen export as a function of mineral substrate (e.g. presumably all the substrates on Archean continents for biological soil crusts were not only sand; if they were, the authors need to discuss why and what evidence there is for this), nor were there any attempts made to account for important variables like water or light intensity in the Archean. Atmospheric pressure and oxygen content were also not accounted for; scarce Archean atmospheric oxygen almost certainly would have influenced the amount of nitrate that was denitrified (likely higher under lower oxygen than today).

Specific comments:

1) It is unclear why the authors bother to specifically mention the Johnston et al. 2007 paper why they then discard its values ("short-term incubations....do not represent a steady-state situation

with respect to production, consumption and diffusion of N species").

2) Figure 1 is not needed; it simply replots data from another paper. It's also unclear what the evidence is for the yellow line "first evidence for BSCs" because there is no reference given and it is not discussed in the text nor in the figure caption.

3) Figure 3: This figure (the only one with data from this study) is confusing because there are two x-axes and it's unclear which one the plotted lines are associated with. What do the dashed vertical lines represent? Also, only France and Europe are shown for % of modern continental area colonized by BSCs, but no references are provided for these values. What about other continents - - additional BSC coverage values must be available or at least approximated for at least the SW US and Israel, and probably for many other countries too. It seems incongruent to use one estimate for N export from Indiana, USA and another for coverage based on a different climate and geological terrain (France).

4) line 150: I believe this should be 0.02 as above, not 0.2 ?

5) The manuscript contains numerous English grammatical mistakes.

RESPONSES TO THE REVIEWERS COMMENTS:

We provide below a thorough answer to questions rose by the two reviewers (their original text is provided first in black followed by our responses in green) and we offer enclosed a deeply revised manuscript. We carefully took into account all the remarks made by the reviewers and provide much more robust estimates including a wider range of published studies on modern BSCs. We also discuss in great details the rationale that we used to transpose these results to the Archean deep time period, including the possible limitation inherent to the use of a modern analog to decipher past mechanisms.

Reviewers' comments:

Reviewer #1 (Remarks to the Author):

We thank the reviewer #1 for her constructive comments and suggestions that significantly improved the manuscript.

- One weakness of the calculations is that they are based on data from only one modern study of BSC (Thiet et al. 2005). However, that paper actually cites several other studies that report similar fluxes (see data on the Colorado Plateau, the Sonoran desert and Australia quoted on page 244). It would strengthen this new study if those values were quoted as well to support the decision of using a N flux of 0.02-0.8 as a reference range.

We fully agree with this comment and discuss N fluxes using estimates available in the literature below, although Thiet's results remains the most directly relevant.

We added direct estimates from Johnson et al., 2007 which measured N short-range export rates from crusts of the Colorado Plateau to subsurface soil ranging from 0.98 to 5.02 g NH₄⁺-N m⁻² yr⁻¹ and 1.10 to 3.8 g NO₃⁻-N m⁻² yr⁻¹ commensurate with rates of N-fixation at the studying site. This measurement has been realized at a depth of 10-20 cm below the crusts in the soil. These estimates are one order of magnitude larger than those of Thiet et al., 2005. However, since considerable fixed N can be subsequently lost in deep soil hot spots via denitrification and volatilization, demonstrating that biological soil crusts fix and release N does not necessarily mean that it is all retained in the system N pool for a long period of time before being efficiently transported in the hydrological system. As such this study does not represent a fully resolved situation with respect to production, consumption and diffusion of N species and thus most probably overestimates the fluxes on a long period of time (notice than in this case denitrification rate was very low, as for much of BSCs studies reports).

Moreover, the authors acknowledge that their short-term incubation (20 hours) experiment do not represent a steady-state situation with respect to production, consumption and diffusion of N species (separate experiments indicated that steady states were not reached as nitrate concentration within the crusts increased continuously during periods longer than the duration of the experiments). In the longer (75 days) experiment of Thiet et al., 2005 N was measured after it had leached through the soil profile beyond a depth of 7 cm. For this reason we think that this estimate reflect a more robust direct estimate of fluxes from long standing exportable N pool in the soil system and that's the reason for which we initially used these numbers.

This said, we could also derived indirect estimates using the N inputs recorded in other ecosystems:

Accordingly,

- biological soil crusts in cold deserts are estimated to fix 1–10 g -N m⁻² yr⁻¹ (Rychert and Skujins, 1974; West and Skujins, 1977).
- Estimates of N inputs by crusts in the Sonoran desert and Australia range from 0.7–1.8 g -N m⁻² yr⁻¹ and 0.13 g -N m⁻² yr⁻¹, respectively (Evans and Johansen, 1999; Rychert et al., 1978). Overall, estimates of annual N inputs from biocrust N fixation are highly variable, ranging from 0.07 g -N m⁻² yr⁻¹ to 10 g -N m⁻² yr⁻¹ [reviewed by Evans and Lange (2001), Belnap (2002a, b), Russow et al. (2005), Stewart et al. (2011a, b, c), and Caputa et al. (2013)].
- Quantification of N fixation by desert biocrusts globally reported an average N-fixation rate of 0.6 g -N m⁻² yr⁻¹ (Elbert et al. 2012).

Once again, these estimates cannot be used directly because considerable fixed N can be lost in the field via biological recycling and volatilization, demonstrating that biological soil crusts fix and release N does not necessarily mean that it is all retained in the soil N pool system.

However, Guo et al., 2008 demonstrate a significant positive correlation between BSCs inorganic nitrogen and topsoil (between 0 and 5cm) inorganic nitrogen concentrations. The relationship demonstrate by this study is that about t ¼ of the inorganic nitrogen present in the crusts finds its way to the underlying topsoil (sand dune in this case). Such estimate is consistent with Kershaw (1985) that reported that 19–28 % of labeled N applied to BSCs was later found in surrounding soils.

Using these indirect estimates and a minimum export rate of 19% we could scale that the minimum export rate of inorganic nitrogen reported in the litterature may vary from around 0.0133 g -N m⁻² yr⁻¹ to 1.9 g -N m⁻² yr⁻¹ with a mean of 0.114 g -N m⁻² yr⁻¹.

This is therefore fully consistent within error to Thiet et al.,'s direct mesurements. All these estimates, direct and indirect, are introduced in **Table 1 and Fig. 3** and discussed in the revised version of the manuscript form **line 114 to line 143**.

Regarding this range, it would be very useful to know what was the main driver for the low and high end, so one can evaluate which values are more common.

The range reported by Thiet et al., 2005 in a given ecosystem is likely to depend on the following environmental parameters: Rainfall intensity, light intensity and leachate volume. Increase in light intensity directly increase the N output fluxes, while decrease in light intensity combined with an increase in rainfall intensity indirectly increase the N output fluxes by maximizing the leachate volume.

Moreover, globally, the export range might also depend on the BSCs ecosystem structure and especially the proportion of N-fixing organisms and their N-fixation rates (Magee and Burris 1954; Silvester et al. 1996; Barger et al., 2016) compared with the proportion and activity of denitrifiers. This question has been addressed by Strauss et al., 2012 using modern examples of desert biological soil crusts across biogeographic regions in the Southwestern United States from the Colorado Plateau and the Mojave, Sonoran and Chihuahuan Deserts. The study shows that the measured rates of major biogeochemical N-transformations (N-fixation, aerobic ammonia oxidation, anammox and denitrification) despite the demonstrable differences in microbial community composition and soil material displayed a remarkably consistent pattern of internal N cycling. Dinitrogen fixation and aerobic ammonia oxidation

were prominent transformations at all sites while, anaerobic ammonia oxidation (anammox) were below the detection limit in all cases, and heterotrophic denitrification was also of little consequence for the flow of N, with rates at least an order of magnitude smaller than those of N-fixation. Environmental parameters as suggested by Thiet et al., 2005 may therefore primarily control the export rate in BSCs over a wide range of microbial community composition and soil material. We discussed this inference in the revised version of the manuscript between **line 148 and line 172**.

- Another concern is whether Precambrian BSCs would indeed have produced as much nitrogen as modern ones. Most of the fixed nitrogen is probably released into soil when the bacterial biofilm is decomposed and cellular organic nitrogen gets released. To what degree is that process dependent on atmospheric O₂ and macrofauna? This may be difficult to assess, but it would be worth adding a comment to the main text about how nitrogen is released from BSCs into the environment.

The notion presented by the reviewer is not applicable to BSCs. There is no need for degradation of the biofilms, all measurement presented in the literature above were done with life, healthy BSC. A broken N cycle and the leaching of cellular N driven by cycles of desiccation and wetting that damage cellular membranes are likely behind the phenomenon. These constraints are likely to have been as strong in the Precambrian as they are now. We added a sentence to discuss the possible fate of released nitrogen from **line 169 to line 172**.

- L. 38-39: Change to “from 1.9 Ga onwards...”. The study by Stüeken (2013, ref. 10) is about the 1.45 Ga Belt Supergroup. You could also add a citation to Koehler et al. (2017, GCA), who reported similar spatial trends in two other basins at 1.5 Ga.

Done

- L. 82: Another relevant reference would be Blank & Sanchez-Barcaldo (2010) who proposed that cyanobacteria first evolved in fresh water.

Done

- L. 89: If you want, you could add a citation to Stüeken et al. (2012, Nature Geoscience), which also concluded oxidative life on land.

Done

- L. 141: The term ‘oxidative weathering’ is misleading. The BSC flux is not a weathering flux, because it does not affect the flux of nitrogen out of rocks. Better change to “... the Phanerozoic N flux from land into the ocean...”. Also change ‘canonical’ to ‘proposed’.

Done

- L. 143, 146: You probably don’t need to call it weight percent (wt. %), if it refers to a surface area?

True thanks for spotting this mistake

- L. 143 and Fig. 3: The 80% value is not shown on the x-axis in Fig. 3. At first, I didn’t know where it came from. It would be good if you could show a small inset in Fig. 3 where the x-axis goes to 100%, such that one can see the intercept with the 0.02 gN/m²/yr flux.

Done

- L. 167: Change to “...Phanerozoic N fluxes from land into the ocean, ...”. As above, the term oxidative weathering is not quite correct here.

Done

- Fig. 1 seems to be missing the new data from Zerkle et al. (2017 Nature) at 2.3 Ga. Since those are

discussed in the text, they would be worth adding. For the Phanerozoic, I recommend adding the database (available online) from Algeo et al. (2014 Biogeosciences)

Done

Reviewer #2 (Remarks to the Author):

We thank the reviewer #2 for her/his constructive comments and suggestions. Her/his comments are in line with reviewer #1 and also question the rationale that we used to transpose these results to the Archean deep time period. We answer below in details point by point to her/his concern and provide a much more robust discussion of the possible pitfalls of our finding. As it is with the used of modern stromatolites as Archean analogues we hope that our study will motivate the BSCs community for addressing new questions with a deep time perspective.

The most major flaw is that the paper's nitrogen export values, which form the basis of the calculations, are from a SINGLE study of BSCs from beach sand dunes on Lake Michigan (Thiet et al., 2005), despite the author's citation of a book chapter (Barger et al., 2016) with section 14.3 ("Nitrogen Release to the Surrounding Substrate") citing numerous (~20) papers on the subject.

See answer above on the same comment from reviewer 1.

Other major problems with the study approach relate to the few variables examined in the calculations. The only factor that was included in the model was fraction of land area for Archean continents. There is no mention of the variability of nitrogen export as a function of mineral substrate (e.g. presumably all the substrates on Archean continents for biological soil crusts were not only sand; if they were, the authors need to discuss why and what evidence there is for this),

In modern examples of desert biological soil crusts across biogeographic regions in the Southwestern United States from the Colorado Plateau and the Mojave, Sonoran and Chihuahuan Deserts, Strauss et al., 2012 show that the measured rates of major biogeochemical N-transformations (N-fixation, aerobic ammonia oxidation, anammox and denitrification) despite the demonstrable differences in microbial community composition and soil material displayed a remarkably consistent pattern of internal N cycling and N export. Dinitrogen fixation and aerobic ammonia oxidation were prominent transformations at all sites while, anaerobic ammonia oxidation (anammox) rates to be below the detection limit in all cases, and heterotrophic denitrification was also of little consequence for the flow of N, with rates at least an order of magnitude smaller than those of N -fixation. **See lines 148-161.** Moreover, it is inferred by a majority of scientists that the early Precambrian (cf. Archaean) was characterized by conditions best described as “weathering aggressive” (Corcoran and Mueller, 2004). These would have been caused by the combination of high levels of heat, humidity, and greenhouse gases like CO₂ and CH₄ (e.g., Pavlov et al., 2001; Thomazo et al., 2009). As a result of this intense Archaean weathering regime and an absence of binding vegetation, labile minerals and unstable rock fragments would have been rapidly broken down to form clay minerals, which were swiftly separated by sediment transport and sorting agents from the predominant quartz and accessory heavy mineral grains more resistant to chemical breakdown forming quartz arenite (i.e. sandstone; Corcoran and Mueller, 2004). Indeed numerous mature to even supermature quartz arenites are known from many ancient cratonic area (Donaldson and de Kemp, 1998). The Precambrian fossil BSC from Arizona’s Dripping Springs Formation is a typical example of such setting (Beraldi et al., 2014). In addition to this the sedimentation has been largely dominated by siliciclastic sedimentation during the

Archean (see Fig. below from Bose et al., 2012) and aeolian sand system is reported as early as 3.2 Ga (Lower Moodies Group of the Swaziland Supergroup in South Africa; Homann et al., 2015). Finally some authors suggested that the specific dynamic configuration of the Precambrian atmosphere (i.e. the density of the 2.7 Ga atmosphere was less than twice modern levels; Som et al., 2012) and its interaction with sediment grains could explain the occurrence of simple but giant transverse dunes with maximum preserved set thicknesses (more than 50 m thick), such as the single aeolian dune cross-bedded set recorded from the Late Neoproterozoic McFadden Formation (Western Australia; Grey et al., 2005).

For all these reasons we do not think that sand substrate might have been a limiting factor to BSCs colonization of the Archean land surface and reasonable estimate of a few tens of percent of BSCs coverage of Archean continent (notice that the nowadays continental coverage is around 16-20%, Elbert et al., 2012; Garcia-Pichel, 2002) seems reasonable.

[Redacted]

Fig. The relative abundance of non-siliciclastic sediments through time, in Bose et al., 2012.

We discussed this inference in the revised version of the manuscript between **line 161 and line 172**.

....nor were there any attempts made to account for important variables like water or light intensity in the Archean.

Increase in light intensity directly increase the N output fluxes, while decrease in light intensity combined with an increase in rainfall intensity indirectly increase the N output fluxes by maximizing the leachate volume according to Thiet et al., 2005. (**lines 144-148**).

There is no ways or studies we are aware of to account for the Archean rainfall intensity and its evolution.

The light intensity of the Archean and its effect on a potential early Earth BSCs community is an interesting question. While the sun was 30% dimmer during the Archean the lack of an ozone layer result in higher irradiance than today (Castenholz & Garcia-Pichel, 2000). However, BSC today are not light-limited in their productivity, literally light intensity often becomes a stress factor. In fact, in modern times, in order to promote the growth of BSC's one has to invariably shade them artificially to obtain optimal results (Velasco-ayuso et al., 2017). A "faint young sun", would have only helped in this regard. We add this inference in the revised version of the manuscript between **line 62 and line 69**.

Atmospheric pressure and oxygen content were also not accounted for; scarce Archean atmospheric oxygen almost certainly would have influenced the amount of nitrate that was denitrified (likely higher under lower oxygen than today).

There are a few estimates of Archean Earth air pressure with reported values fluctuating from 0.52 to 1.1 bar from 2.73Gyr fossil raindrop (Som et al., 2012), whereas isotopic studies on 3.0-3.5Gyr rocks suggest an upper limit of 0.5-1.2 bar (Marty et al., 2013) and pressure calculated from subaerial lava flows suggests that the sea-level air pressure at 2.74Gyr was 0.23 ± 0.23 bar (Som et al., 2016). Following these numbers one can postulate that the early nitrogen cycle, N-fixation for instance, may have proceed with different kinetic rate (faster) than nowadays and that such differences may weaken our estimates. We acknowledge that this is one limit of our approach at **line 200** and that more work, beyond the scope of this paper, is needed to address such a complex question.

It is well established that before 2.4 Ga the archean environment was anoxic with atmospheric pO_2 below 10^{-5} the present atmospheric level (Pavlov and Kasting, 2002). The existence of oxygen oases in restricted ecosystem (e.g. lake, bound) is nonetheless often argued to explain occurrence of redox cycling of elements (including nitrogen but also, sulfur, carbon and metallic elements; Lalonde & Konhauser, 2015). On the geological time scale, in the Archean environment the overall low O_2 concentration might have indeed triggered/enhanced removal of nitrate from the hydrological system (early close to production site or later in the ocean) to the atmosphere by denitrification. This notion is acknowledged and discussed from **lines 228 to 231** (already in the initial version of this manuscript). Our point being that BSCs release of fixed-N might have been a driving force to the evolution of denitrifiers and a complete N cycle. We also note that the formation of nitrate from ammonium in the crusts is powered by the oxygen locally produced by cyanobacteria, and that this process would have occurred even if the atmosphere was anoxic. On a shorter biological time scale and according to Johnson et al., 2007 if denitrification is expected to proceed in BSCs community it is however not efficient even under low O_2 with rates at least an order of magnitude smaller than those of N-fixation perhaps (but not surely) due to limited C availability for heterotrophic respiration. In conclusion, it is difficult to speculate on how efficient denitrification was in the Archean environment on biological short time scale but we can postulate that the accumulation of nitrate and ammonium in their associated hydrological system is a reasonable hypothesis. We rewrite the discussion paragraph associated with this question (between **line 200 and line 214**)

Moreover, in order to provide future guide line, we added a paragraph that predict isotopic signature that might be imprint in the geological record of Archean fossil BSCs (such as the one reported in the Moodies Group) in order to testify the fate of nitrate produced by these ecosystem isotopically. Indeed, if all produced nitrate is quantitatively denitrified the $\delta^{15}N$ signature of BSCs OM might be close to 0‰ (the atmospheric value of N_2), while if the nitrate pool is not quantitatively denitrified we could expect a positive BSCs $\delta^{15}N$ record. These perspectives have been added to the manuscript from **lines 215 to 219**.

Specific comments:

1) It is unclear why the authors bother to specifically mention the Johnston et al. 2007 paper why they then discard its values ("short-term incubations....do not represent a steady-state situation with respect to production, consumption and diffusion of N species").

This is now better explained in the manuscript from **line 124 to line 132** and shown on **Fig. 3 and Table 1**.

2) Figure 1 is not needed; it simply replots data from another paper. It's also unclear what the evidence is

for the yellow line "first evidence for BSCs" because there is no reference given and it is not discussed in the text nor in the figure caption.

This paper is clearly a contribution for a broad audience including more than stable isotope geochemist and Precambrian geologist but also ecologist and microbiologist that are not necessarily familiar with nor the nitrogen isotope secular variations through the Precambrian, neither the deep time evolution of the nitrogen biogeochemical cycling – it is optimal in our opinion to show this diagram and give the associated reading line in the text. We think that the expectedly broad audience interested by this contribution will appreciate this state of the art picture of biogeochemical cycle evolution interpretations using $\delta^{15}\text{N}$ secular variations.

The reference has been included in the **Fig. 1**.

3) Figure 3: This figure (the only one with data from this study) is confusing because there are two x-axes and it's unclear which one the plotted lines are associated with. What do the dashed vertical lines represent? Also, only France and Europe are shown for % of modern continental area colonized by BSCs, but no references are provided for these values. What about other continents -- additional BSC coverage values must be available or at least approximated for at least the SW US and Israel, and probably for many other countries too. It seems incongruent to use one estimate for N export from Indiana, USA and another for coverage based on a different climate and geological terrain (France).

The Figure 3 has been completely redrawn – it is including more estimates of N export from direct and indirect measurements taken from an exhaustive compilation of previous studies. We removed dashed vertical lines and better explain that France and Europe was indicated on the X axis only to provide a means of understanding of the area that a given % represents, i.e. 7% means an area that is close to the European continental size. We realize that our explanations were not clear enough, and confusing and we clarify the figure caption accordingly.

4) line 150: I believe this should be 0.02 as above, not 0.2 ?

In the initial manuscript it was 0.2 as we wanted to illustrate what estimates of colonization gives a moderate flux. Since we include many more estimates in the revised manuscript we make sure to avoid this kind of confusion now. The new Figure 3 with inset (x-axis up to 100%), also clarifies this by showing the intercept of reported export rates with the Phanerozoic N flux from land into the ocean ($0.15 \text{ Tmol yr}^{-1}$ on the Y axis).

5) The manuscript contains numerous English grammatical mistakes.
The grammar has been carefully revised.

New References:

Rychert R C, Skukins J, Sorensen D and Porcella D (1978) Nitrogen fixation by lichens and free-living microorganisms in deserts. In: Nitrogen in Desert Ecosystems Dowden. Ed. N E J J West Skujins pp. 20–30. Hutchinson, and Ross, Inc, Stroudsburg.

West N E and Skujins J (1977) The nitrogen cycle in North American cold-winter semi-desert ecosystems. *Oecologia* 12, 45–53.

Evans R D and Johansen J R (1999) Microbiotic crusts and ecosystem processes. *Crit. Rev. Plant Sci.* 18, 183–225.

Rychert R C, Skukins J, Sorensen D and Porcella D (1978) Nitrogen fixation by lichens and free-living microorganisms in deserts. In: Nitrogen in Desert Ecosystems Dowden. Ed. N E J J West Skujins pp. 20–30. Hutchinson, and Ross, Inc, Stroudsburg.

Evans R D and Lange O L (2001) Biological soil crusts and ecosystem nitrogen and carbon dynamics. In: Biological Soil Crusts: Structure, Function, and Management, Ecological Studies, Vol. 150. Eds. J Belnap and O L Lange pp. 264–279. Springer-Verlag, Berlin.

Belnap J (2002a) Nitrogen fixation in biological soil crusts from southeast Utah, USA. *Biol Fertil Soils* 35:128–135.

Belnap J (2002b) Impacts of off road vehicles on nitrogen cycles in biological soil crusts: resistance in different US deserts. *J Arid Environ* 52:155–165.

Russow R, Veste M, Böhme F (2005) A natural ^{15}N approach to determine the biological fixation of atmospheric nitrogen by biological soil crusts of the Negev Desert. *Rapid Commun Mass Spectrom* 19:3451–3456.

Stewart KJ, Coxson D, Grogan P (2011a) Nitrogen inputs by associative cyanobacteria across a low arctic tundra landscape. *Arct Antarct Alp Res* 43:267–278.

Stewart KJ, Coxson D, Siciliano SD (2011b) Small-scale spatial patterns in N_2 -fixation and nutrient availability in an arctic hummock–hollow ecosystem. *Soil Biol Biochem* 43:133–140.

Stewart KJ, Lamb EG, Coxson DS, Siciliano SD (2011c) Bryophyte-cyanobacterial associations as a key factor in N_2 -fixation across the Canadian Arctic. *Plant Soil* 344:335–346.

Caputa K, Coxson D, Sanborn P (2013) Seasonal patterns of nitrogen fixation in biological soil crusts from British Columbia's Chilcotin grasslands. *Botany* 641:631–641.

Elbert W, Weber B, Burrows S, Steinkamp J, Budel B, Andreae MO, Poschl U (2012) Contribution of cryptogamic covers to the global cycles of carbon and nitrogen. *Nat Geosci* 5:459–462.

Guo Y, Zhao H, Zuo X, Drake S, Zhao X (2008) Biological soil crust development and its topsoil properties in the process of dune stabilization, Inner Mongolia, China. *Environ Geol* 54:653–662.

Kershaw KA (1985) Physiological ecology of lichens. Cambridge University Press, London.

Magee WE, Burris RH (1954) Fixation of N_2 and utilization of combined nitrogen by *Nostoc muscorum*. *Am J Bot* 41:777–782.

Silvester WB, Parsons R, Watt PW (1996) Direct measurement of release and assimilation of ammonia in the *Gunnnera-Nostoc* symbiosis. *New Phytol* 132:617–625.

Strauss SL, Day TA, Garcia-pichel F (2012) Nitrogen cycling in desert biological soil crusts across biogeographic regions in the Southwestern United States. *Biogeochemistry* 108:171–182.

Corcoran, P.L., Mueller, W.U. (2004) Archaean sedimentary sequences. In: Eriksson, p.g., Altermann, W., Nelson, D.R., Mueller, W.U., Catuneanu, O. (Eds.) *The Precambrian Earth: tempos and events*. Elsevier, Amsterdam, pp. 613-625.

Pavlov, A. A., Kasting, J. F. & Brown, L. L. (2001) UV-shielding of NH_3 and O_2 by organic hazes in the Archean atmosphere. *J. Geophys. Res.* 106, 23 267–23 287.

Donaldson, J.A., de Kemp, E.A. (1998) Archean quartz arenites in the Canadian Shield: examples from the superior and Churchill Provinces. *Sedimentary Geology* 120, 153–176.

Bose, P.K., Eriksson, P.G., Sarkar, S., Wright, P., Samanta, P., Mukhopadhyay, S., Mandal, S., Banerjee, S., Altermann, W. (2012) Sedimentation patterns during the Precambrian: a unique record? *Marine and Petroleum Geology* 33, 34–68.

Som, S. M., Catling, D. C., Harnmeijer, J. P., Polivka, P. M. & Buick, R. (2012) Air density 2.7 billion years ago limited to less than twice modern levels by fossil raindrop imprints. *Nature* 484, 359–362.

Grey, K., Hocking, R.M., Stevens, M.K., Bagas, I., Carlsen, G.M., Irimies, F., Pirajno, F., Haines, P.W. and Apak, S.N. (2005) Lithostratigraphic nomenclature of the Officer Basin and correlative parts of the Paterson Orogen, Western Australia. Western Australia Geological Survey, Report 93, 89 pp.

Castenholz, R.W., Garcia-Pichel, F. (2000) Cyanobacterial responses to UV-radiation. In: Whitton, B.A., Potts, M. (Eds.), *The Ecology of Cyanobacteria*. Kluwer Academic Publishers, Dordrecht, pp. 591-611.

Marty, B., Zimmermann, L., Pujol, M., Burgess, R. & Philippot, P. (2013) Nitrogen isotopic composition and density of the Archean atmosphere. *Science* 342, 101–104.

Som, S.M., Buick, R., Hagadorn, J.W., Blake, T.S., Perreault, J.M., Harnmeijer, J.P. and Catling, D. (2016) Earth's air pressure 2.7 billion years ago constrained to less than half of modern levels. *Nature Geoscience*: DOI: 10.1038/NGEO2713.

Pavlov, A. A. & Kasting, J. F. (2002) Mass-independent fractionation of sulfur isotopes in Archean sediments: Strong evidence for an anoxic Archean atmosphere. *Astrobiology* 2, 27–41.

Velasco Ayuso S., Giraldo Silva A., Nelson C.J. Barger NN, and **Garcia-Pichel F** (2017) Microbial nursery production of high-quality biological soil crust biomass for restoration of degraded dryland soils. *Appl. Environ. Microbiol* 83: 3 e02179-16.

Reviewers' comments:

Reviewer #1 (Remarks to the Author):

Review of "Possible nitrogen fertilization of the early Earth ocean by microbial continental ecosystems" by Thomazo et al.

The authors have invested considerable effort into addressing all the major points raised in the previous rounds of reviews. The manuscript makes a convincing case for the importance of biological soil crusts (BSCs) in the ancient nitrogen cycle. It is a provocative contribution that is likely to trigger significant follow-up research.

A few minor comments that can be addressed before publication:

I. 16: I suggest changing 'cyanobacteria' to just 'bacterial'. There is still a lot of controversy in the literature about when cyanobacteria first arose. Even though it is quite possible that they existed in the mid-Archean, the effects of biological soil crusts should also apply if they were composed of other organisms.

I. 37: change 'became available' to 'became more widely available'. This would be more accurate because the oxygen oases in the Archean may already have contained some bioavailable nitrate.

I. 38: change 'Paleoproterozoic' to 'mid-Proterozoic'

I. 64: should 'literally light intensity' be changed to 'high light intensity'?

I. 199: It would perhaps be worth adding a sentence here to say explicitly that this flux is not equal to weathering. Otherwise, it might be misquoted in the literature one day. I suggest something like the following sentence: "Importantly, this flux is not equal to the weathering flux of N contained in continental crust, because BSCs with diazotrophic organisms are capable of fixing their own N directly from the atmosphere."

I. 218-220: I would suggest removing these isotopic predictions. A value of zero permil could also simply reflect an anaerobic BSC with little nitrate production. It would not rule out N export to the ocean as ammonium.

I. 229 onwards: It is quite possible that early BSCs (with or without cyanobacteria) underneath an anoxic atmosphere were primarily exporting ammonium rather than nitrate. So I suggest changing the wording in the last two paragraphs with perhaps less emphasis on nitrate.

I look forward to citing the paper.

Best regards,
Eva Stüeken

Reviewer #3 (Remarks to the Author):

This study intends to propose and evaluate an interesting hypothesis that the export of fixed nitrogen from terrestrial biological soil crusts (BSCs) to marine systems would have been ultimately an important source of new N for the ocean.

To effectively evaluate their proposal, the authors must consider i) whether there was substantial N export from BSCs to the underlying Archean topsoil, ii) whether topsoil N was effectively transported to the oceans without removal through other pathways, and iii) whether the coverage of BSCs was large enough to scale up total N export by BSCs to a globally significant value. The

authors use N export studies of modern BSCs as the basis of evaluation.

Each of the previous reviewers has raised questions regarding the reliability of using a single study (Thiet 2005) to draw the main conclusion that BSCs, covering only a few percent of land mass, were important contributors to marine N. While it appears that the revised ms has responded to this criticism, by increasing the number of studies included Fig 3, the authors continue to focus on using the Thiet study's highest value of export (0.8) to draw the main conclusion without adequate justification. Considering Archean hydrological cycle, rainfall, is essential in justifying using moderate to high N export values as the basis for calculations, but this was not addressed in the revised ms. As it stands, using more conservative N export values from Thiet would result in BSCS having to cover tens of percent to >100% of land, a result that is perhaps less dramatic than 3% land mass colonization. In particular, the abstract sentence "if as little as 3% of the Archean continent were colonized by BSCs with biogeochemical properties similar to those seen today, the net output flux of inorganic N reaching the Precambrian hydrogeological system would have been of the same order of magnitude as that of modern continent" is somewhat misleading. A more conservative estimation would lead to 0.005 – 0.2 Tmol.yr⁻¹ (for a 3% land mass colonization), which, while encompassing the modern value of N export 0.15 Tmol.yr⁻¹, varies across several magnitudes.

A secondary criticism of both reviewers lies in the difficulty in relating the factors driving current estimation of N export with the conditions that would have taken place on the early Earth. The authors adequately respond to issues related to soil substrate. However, the impact of atmospheric pressure, N partial pressure, and other conditions while acknowledged is not fully addressed.

Another gap in this study is consideration of how much topsoil N is transported to the ocean, particularly if land mass colonization were moderate or large, not limited to coasts, as would be expected if BSCS exported low to moderate amounts of N. What is the fate of this N after the topsoil step? Is there evidence of BSC N transfer to close or distant water bodies? This needs to be clarified.

Additionally, some replies to reviewers on the effect of grazers and light intensity were poorly supported. Literature exists on the effect of grazers on cyanobacteria in BSCs, with contrasting effect on the impact of precipitation (Ghabbour et al 1980, Grazing by microfauna and productivity of heterocystous nitrogen-fixing blue-green algae in desert soils, *Oikos* 34: 209-218). The discussion of light intensity would benefit from a better separation of the effect of light limitation from Photosynthetic Active Radiation (PAR) and UV radiation. The faint sun during Archean time produced less PAR, but because of the lack of ozone, more UV light could be expected to be transmitted. Thus Archean environment might have both been light limiting, and UV damaging. While this might not preclude biological life on earth (Cockell and Horneck 2001), it would not have "help in this regards", as stated in the response to reviewer 2 comments. All those aspects would probably not change the outcome of the paper, but it would result in a better discussion.

Minor comments:

- 1) The word "from" is often misspelled "form"
- 2) The wt.% has not been changed in the revised version (see 8th comment from reviewer)
- 3) Table 1 - meant for SI or in main ms?
- 4) Fig 3 remains difficult to understand - the yellow and green background shading is unnecessary if readers should focus attention on dotted lines

RESPONSES TO THE REVIEWERS COMMENTS:

We provide below a thorough answer to questions raised by the two reviewers #1 and #3 (their original text is provided first in black followed by our responses in green). We carefully took into account all the remarks made by the reviewers and are enclosing a revised version of our manuscript implementing these changes. We also further discuss the rationale that we used to transpose these results to the Archean deep time period.

Reviewer #1 (Remarks to the Author): The authors have invested considerable effort into addressing all the major points raised in the previous rounds of reviews. The manuscript makes a convincing case for the importance of biological soil crusts (BSCs) in the ancient nitrogen cycle. It is a provocative contribution that is likely to trigger significant follow-up research.

We thank the reviewer #1 for her enthusiastic comments.

A few minor comments that can be addressed before publication:

I. 16: I suggest changing 'cyanobacteria' to just 'bacterial'. There is still a lot of controversy in the literature about when cyanobacteria first arose. Even though it is quite possible that they existed in the mid-Archean, the effects of biological soil crusts should also apply if they were composed of other organisms.

Done

I. 37: change 'became available' to 'became more widely available'. This would be more accurate because the oxygen oases in the Archean may already have contained some bioavailable nitrate.

Done

I. 38: change 'Paleoproterozoic' to 'mid-Proterozoic'

Done

I. 64: should 'literally light intensity' be changed to 'high light intensity'?

Right! Done

I. 199: It would perhaps be worth adding a sentence here to say explicitly that this flux is not equal to weathering. Otherwise, it might be misquoted in the literature one day. I suggest something like the following sentence: "Importantly, this flux is not equal to the weathering flux of N contained in continental crust, because BSCs with diazotrophic organisms are capable of fixing their own N directly from the atmosphere."

Indeed thanks for this addition. We introduce the sentence line 216.

I. 218-220: I would suggest removing these isotopic predictions. A value of zero permil could also simply reflect an anaerobic BSC with little nitrate production. It would not rule out N export to the ocean as ammonium.

That is true. We removed these predictions.

I. 229 onwards: It is quite possible that early BSCs (with or without cyanobacteria) underneath an anoxic atmosphere were primarily exporting ammonium rather than nitrate. So I suggest changing the wording in the last two paragraphs with perhaps less emphasis on nitrate.

We modified the paragraph as follow: "The delivery of nitrate **and ammonium** to the hydrogeological reservoir would also have provided a new evolutionary driver and an opportunity to trigger diversification of prokaryotes and eukaryotic phytoplankton **able to use fixed nitrogen as a source of nutrient** before nitrate concentrations stabilized in the ocean at around 2.3 Ga."

Reviewer #3 (Remarks to the Author):

This study intends to propose and evaluate an interesting hypothesis that the export of fixed nitrogen from terrestrial biological soil crusts (BSCs) to marine systems would have been ultimately an important source of new N for the ocean.

To effectively evaluate their proposal, the authors must consider i) whether there was substantial N export from BSCs to the underlying Archean topsoil, ii) whether topsoil N was effectively transported to the oceans without removal through other pathways, and iii) whether the coverage of BSCs was large enough to scale up total N export by BSCs to a globally significant value. The authors use N export studies of modern BSCs as the basis of evaluation.

We thank the reviewer #3 for her/his constructive comments and suggestions that significantly improved the manuscript. See our point-by-point answer below.

Each of the previous reviewers has raised questions regarding the reliability of using a single study (Thiet 2005) to draw the main conclusion that BSCs, covering only a few percent of land mass, were important contributors to marine N. While it appears that the revised ms has responded to this criticism, by increasing the number of studies included Fig 3, the authors continue to focus on using the Thiet study's highest value of export (0.8) to draw the main conclusion without adequate justification.

We have taken a great care in the revised manuscript to remove the emphasis from Thiet estimates and highlight more conservative estimates. For this purpose we revised the abstract (see comment below) but also revised line 212: "In both modern and Archean configurations, a high to moderate range of net N outputs from BSCs to subsurface soil of 0.8 to 0.1 g N m⁻² yr⁻¹ (with an average indirect estimates of 0.114 g N m⁻² yr⁻¹ ⁴⁰) causes a total inorganic N transport comparable to Phanerozoic N fluxes from land into the ocean at small value (≈3 to 30% based on Thiet et al.³⁷ direct estimates and 20% based on Elbert et al.⁴⁰ global indirect average estimate) of colonization (Fig. 3)."

Considering Archean hydrological cycle, rainfall, is essential in justifying using moderate to high N export values as the basis for calculations, but this was not addressed in the revised ms.

To the extent of our knowledge, the Archean rainfall regime is not constrained, and we do not imply that it was more intense. We agree with the reviewer that putting forward the highest value from Thiet et al. was not the most conservative option and we have now revised the manuscript to highlight the average value from Elbert et al.

As it stands, using more conservative N export values from Thiet would result in BSCs having to cover tens of percent to >100% of land, a result that is perhaps less dramatic than 3% land mass colonization. In particular, the abstract sentence "if as little as 3% of the Archean continent were colonized by BSCs with biogeochemical properties similar to those seen today, the net output flux of inorganic N reaching the Precambrian hydrogeological system would have been of the same order of magnitude as that of modern continent" is somewhat misleading. A more conservative estimation would lead to 0.005 – 0.2 Tmol.yr⁻¹ (for a 3% land mass colonization), which, while encompassing the modern value of N export 0.15 Tmol.yr⁻¹, varies across several magnitudes.

We thank the reviewer for pointing this out. We modified the sentence in the abstract in order to temper our point as suggested. See line 20: "Here we show that, if landmasses were colonized by BSCs with biogeochemical properties similar to those seen today, the net output flux of inorganic N reaching the Precambrian hydrogeological system would have been of the same order of magnitude as that of modern continents for a range of inhabited surface area varying from around thirty to as little as a few percent."

A secondary criticism of both reviewers lies in the difficulty in relating the factors driving current estimation of N export with the conditions that would have taken place on the early Earth. The authors adequately respond to issues related to soil substrate. However, the impact of atmospheric pressure, N partial pressure, and other conditions while acknowledged is not fully addressed.

We thank the reviewer for pointing this out. As mentioned in the first round of review estimates of these parameters are scarce in the literature and vary widely among authors. The estimates of Archean Earth air pressure range from 0.52 to 1.1 bar from 2.73 Gyr fossil raindrop evidence (Som et al., 2012), whereas isotopic studies (nitrogen and argon isotopes) on 3.0-3.5 Gyr rocks suggest an upper limit of 0.5-1.2 bar for the N₂ partial pressure (Marty et al., 2013) and pressure calculated from subaerial lava flows suggests that the sea-level air pressure at 2.74 Gyr was 0.23 ± 0.23 bar (Som et al., 2016). Because atmospheric chemistry is largely dominated by N₂ estimates of Archean air pressure and partial pressure of N₂ are congruent. However, even if significant atmospheric pN₂ swings on Earth are proven to be true, it is unlikely that, even when taking into account the most dramatic estimates of ½ of present di-nitrogen in the atmosphere, nitrogen become a limiting factor to microbial primary productivity. Low dissolved nitrogen concentration, has not been reported as a limiting factor to biological nitrogen fixation, unlike molybdenum cofactors, and the reactions seem to be quasi first order down to very low pN₂, in the mbar range (Klinger et al., 1989). Furthermore, it has been already demonstrated that microbial nitrogen fixation evolved very early in Earth's history and that a nitrogen crisis for the primordial biosphere is not evidenced in the geological record (Stüeken et al., 2015; Weiss et al., 2016). We added a paragraph starting line 222 to provide our reader more context regarding the current level of knowledge and approximation of these parameters: "The Archean Earth air pressure estimates range from 0.52 to 1.1 bar based on 2.73 Ga fossil raindrop imprint⁶⁴, whereas nitrogen and argon isotopic studies suggest an upper limit of 0.5-1.2 bar for the N₂ partial pressure at around 3.0 to 3.5 Ga⁶⁵ and pressure calculated from subaerial lava flows suggests that the sea-level air pressure at 2.74 Ga was 0.23 ± 0.23 bar¹⁵. However, even if significant atmospheric pN₂ swings on the early Earth are suggested it is unlikely that nitrogen became a limiting factor for microbial primary productivity^{3,5}. Future works are nevertheless needed to better-characterized variations in the kinetic rate of N-fixation with changing pressure-temperature parameters."

Questions related to the oxygen content has been already addressed in great details in the first round of revisions (from lines 229 to 240).

Another gap in this study is consideration of how much topsoil N is transported to the ocean, particularly if land mass colonization were moderate or large, not limited to coasts, as would be expected if BSCS exported low to moderate amounts of N. What is the fate of this N after the topsoil step? Is there evidence of BSC N transfer to close or distant water bodies? This needs to be clarified.

The fate of N after the topsoil step is not well documented in the literature. However, it has been shown that large reservoirs of nitrate accumulate in the subsoil vadose zone of non-riparian arid environment for thousand of years (Walvoord et al., 2003; sentence added line 126). In riparian system the nitrate will be transported in the groundwater of the watershed to rivers, estuaries, and other coastal waters through the hydrological gradient. This long-range N exports through the hydrological gradient are well established today. There is no reason to expect that the geophysical drivers of this process would have been different in the Archaeal (see for example, Yang et al., 2015) and according to the time scale of interest of this study (i.e. the evolution of the N cycle over hundreds of millions of year) it seems reasonable to

hypothesize that the nitrate released in the soil *sensu largo* will ultimately be delivered to the ocean.

Additionally, some replies to reviewers on the effect of grazers and light intensity were poorly supported. Literature exists on the effect of grazers on cyanobacteria in BSCs, with contrasting effect on the impact of precipitation (Ghabbour et al 1980, Grazing by microfauna and productivity of heterocystous nitrogen-fixing blue-green algae in desert soils, *Oikos* 34: 209-218).

As far as we know there is no grazers (even in the form of protists) known during the Archean time period. Eukaryotic cells are suggested to develop around 1.6–2.1 Ga (e.g. Knoll et al., 2006). This is mentioned line 88 of the manuscript.

The discussion of light intensity would benefit from a better separation of the effect of light limitation from Photosynthetic Active Radiation (PAR) and UV radiation. The faint sun during Archean time produced less PAR, but because of the lack of ozone, more UV light could be expected to be transmitted. Thus Archean environment might have both been light limiting, and UV damaging. While this might not preclude biological life on earth (Cockell and Horneck 2001), it would not have “help in this regards”, as stated in the response to reviewer 2 comments. All those aspects would probably not change the outcome of the paper, but it would result in a better discussion.

We largely revised and extended the discussion on this inference and propose a new paragraph from line 63-84 that includes the reviewer concern:

“During the Archean the lack of an ozone layer resulted in higher irradiance than today despite the fact that the sun was 30% dimmer¹⁹. Because of this peculiar environmental condition Berkner & Marshall (1965) first postulated that the colonization of the landmasses was not possible before the formation of an ozone shield. However, recent findings demonstrate that the peculiar chemistry of the Archean atmosphere may substantially attenuated UV radiation because in the primitive anoxic atmosphere sulfur vapor composed of sulfur molecules and hydrocarbon smog may have strongly screen ultraviolet radiation (Kasting et al., 1989). Also, a high concentration of ferrous ion (Fe II) may have been present in anoxic waters to significantly screen UV radiation (Pierson, 1994). Moreover, Cockell & Raven (2007) show that under a worst-case UV flux (no environmental UV screen) on the Archean Earth, the landmasses could have been colonized by early photosynthesizers. Assuming repair processes similar to organisms on the present-day Earth, organisms capable of tolerating the UV flux found on the exposed surface of the present-day Earth, there would have been zones in a diversity of substrates including sandstone in which phototrophs would be exposed to a UV flux similar to the surface of the present-day Earth, but where photosynthetically active radiation would still be sufficient for photosynthesis. Moreover, modern BSC are not light-limited in their productivity and high light intensity often becomes a stress factor. In fact, in modern times, in order to promote the growth of BSC’s one has to invariably shade them artificially to obtain optimal results²⁰. Therefore, despite a “faint young sun” and the absence of a UV-protective ozone layer in the Archean^{21,22} a terrestrial phototrophic biosphere composed of systems similar to modern BSCs may have existed early before the GOE²³ and could have largely colonized the exposed land surfaces^{24,25}.”

Minor comments:

1) The word “from” is often misspelled “form”

Thanks for spotting this edit. We carefully checked the manuscript.

2) The wt.% has not been changed in the revised version (see 8th comment from reviewer

Done

3) Table 1 - meant for SI or in main ms?

We might have it in the main manuscript if it meets editorial constraints.

4) Fig 3 remains difficult to understand - the yellow and green background shading is unnecessary if readers should focus attention on dotted lines.

Shaded area represent range of values while dotted line represent either mean or peak values of a given range. We thought having the two information on the Figure is worth of interest for a rapid visual understanding of how diverse are estimates form the literature.

Additional references:

Hao, J., Sverjensky, D.A. & Hazen, R.M. (2017) A model for late Archean chemical weathering and world average river water. *Earth Planet. Sci. Lett.* **457**, 191–203.

Klingler, J. M., Mancinelli, R. L. and White, M. R. (1989) Biological Nitrogen Fixation under Primordial Martian Partial Pressures of Dinitrogen, *Adv. Space Res.* **9**, 173–176.

Knoll, A., Javaux, E., Hewitt, D. & Cohen, P. (2006) Eukaryotic organisms in Proterozoic oceans. *Philosophical Transactions of the Royal Society B.* **361** (1470): 1023–38.

Kasting, J. F., Zahnle, K. J., Pinto, J. P. & Young, A. T. (1989) Sulfur, ultraviolet radiation and the early evolution of life. *Orig. Life Evol. Biosph.* **19**, 95–108.

Berkner, L. V. & Marshall, L. C. (1965) History of major atmospheric components. *Proc. Natl Acad. Sci. USA* **53**, 1215–1225.

Pierson BK (1994) The emergence, diversification, and role of photosynthetic eubacteria. In: Bengtson S, Bergström J, Vidal G, Knoll A (eds) Early life on earth. Columbia University Press, New York, pp 161–180, 605 pp.

Cockell, C.S. & Raven, J.A. (2007) Ozone and life on the early Earth. *Philos. Transact. A Math. Phys. Eng. Sci.* **365**, 1889–1901.

Yang, Q., Tian, H., Friedrichs, M. A. M., Hopkinson, C. S., Lu, C., & Najjar, R. G. (2015) Increased nitrogen export from eastern North America to the Atlantic Ocean due to climatic and anthropogenic changes during 1901–2008, *J. Geophys. Res.-Biogeo.*, **120**, 1046–1068.

Walvoord, M. A., F. M. Phillips, D. A. Stonestrom, R. D. Evans, P. C. Hartsough, B. D. Newman, and R. G. Striegl. (2003) A reservoir of nitrate beneath desert soils. *Science*, **302**, 1021–1024.

Reviewers' comments:

Reviewer #3 (Remarks to the Author):

The authors have improved the manuscript. However, the main findings of the study remain highly hypothetical due to the numerous uncertainties in the various components of their model, from the estimate of N fixation activity, percentage of export and more importantly the coverage of land by BSCs needed to obtain substantial contribution. Most of the inferences made remain extremely hard to constrain, and result in high variation in the possible outputs (from 3% to more than 100% of coverage needed to obtain modern time export flux). Furthermore, it is not clear how land to ocean N transport compares with N input directly into the ocean from nitrogen fixation and lightening etc. This is critical in assessing whether BSCs contribute significant amounts of N to the marine systems. In sum, the hypothesis remains interesting, but given all the uncertainties, the work is more suitable for publication in a field specific journal.

Minor comments:

Table 1 is hard to read and redundant to Figure 3. It represents the decomposition of the calculation literature value with the formula from M&M. The main interest of the study is to constrain the hypothesis that BSCs could have contributed to N cycle in the Pre-cambrian. One important unknown factor is the surface coverage that allow similar value than in modern time. A simpler table indicating the percentage coverage to which the estimate based on the different literature reaches the modern value for both Precambrian and modern time would be far more informative and less hard to read.

I was not able to find the value of $0.6 \text{ g-N.m}^2.\text{yr}^{-1}$ used in the manuscript in reference to Elbert et al 2012. The value found in the main manuscript and in Sup. Info. for Cryptogamic ground cover in desert ecosystems was established at $0.76 \text{ g.m}^2.\text{yr}^{-1}$. Was the value of $0.6 \text{ g-N.m}^2.\text{yr}^{-1}$ recalculated from a subset of the literature in the Sup. Info Table S11 of Elbert et al? Elbert et al 2012 encompasses numerous references with lichen and mosses which could indeed be used for this estimation, but it is important to explain which study has been incorporated based on specific criteria and what indicator of centrality (geometric mean, arithmetic mean, ...) has been used.

RESPONSES TO THE REVIEWERS COMMENTS:

We provide below a thorough answer to questions raised by the reviewer #3 (their original text is provided first in black followed by our responses in green). We carefully took into account all the remarks made by the reviewer and are enclosing a revised version of our manuscript implementing these changes.

Reviewer #3 (Remarks to the Author):

The authors have improved the manuscript. However, the main findings of the study remain highly hypothetical due to the numerous uncertainties in the various components of their model, from the estimate of N fixation activity, percentage of export and more importantly the coverage of land by BSCs needed to obtain substantial contribution. Most of the inferences made remain extremely hard to constrain, and result in high variation in the possible outputs (from 3% to more than 100% of coverage needed to obtain modern time export flux).

Although we acknowledge that the range of values that we present is wide due to the high variability of the available data among studies, the limited numbers of analysis of each report does not allow us to perform further descriptive statistical analyses. For these reasons we decided to display all values including extreme values. In order to provide to the reader a clear summary of the range of calculated results we now provide a table that compiles all the literature data and the calculations we derived from them. The geometric mean value given globally by Elbert et al., which is probably the more statistically robust available estimate suggests 17% colonization, consistent with the ranges calculated from the Sonoran (from 7 to 18%) and cold (from 1 to 12 %) deserts datasets.

We completely agree with the reviewer#3 that we did not stress out that point enough in the previous version of our manuscript and include the new table 1 and associated discussion, accordingly.

Furthermore, it is not clear how land to ocean N transport compares with N input directly into the ocean from nitrogen fixation and lightening etc. This is critical in assessing whether BSCs contribute significant amounts of N to the marine systems.

The flux of nitrogen fixed by lightning is orders of magnitude lower than the fluxes discussed here, Navarro-Gonzalez et al. estimated it to be inferior to 0.021 Tmol N/yr. (Navarro-González, R., McKay, C. P. & Mvondo, D. N. A possible nitrogen crisis for Archean life due to reduced nitrogen fixation by lightning. *Nature* 412, 6164 (2001)).

Reference added line 237 of the revised manuscript.

On the modern Earth, rates of nitrogen fixation are nearly equally balanced between continents and oceans (Canfield, D. E., Glazer, A. N., & Falkowski, P. G. (2010). The evolution and future of Earth's nitrogen cycle. *science*, 330(6001), 192-196.) Assuming that this balance also prevailed at the Archean time and given that 19-28% of continental N fixed by biocrust is exported (see line 137 of the manuscript) one could speculate that the land to ocean N transport could be up to 28% of the oceanic nitrogen fixation value. Although very interesting, we feel like the nitrogen cycle and its balance between continent and ocean are not constrained enough for the Archean period to be able to ascertain the value of this hypothesis, we will therefore not include this point in the revised manuscript.

In sum, the hypothesis remains interesting, but given all the uncertainties, the work is more suitable for publication in a field specific journal.

We agree with the reviewer that our work present an interesting hypothesis, that even if grounded with all available currently available data remains speculative. We therefore will make sure to revise the manuscript in order to make it very clear that this manuscript presents a hypothesis that will trigger interesting work but no assertive conclusion.

Minor comments:

Table 1 is hard to read and redundant to Figure 3. It represents the decomposition of the calculation literature value with the formula from M&M. The main interest of the study is to constrain the hypothesis that BSCs could have contributed to N cycle in the Pre-cambrian. One important unknown factor is the surface coverage that allow similar value than in modern time. A simpler table indicating the percentage coverage to which the estimate based on the different literature reaches the modern value for both Precambrian and modern time would be far more informative and less hard to read.

We moved the table 1 in supplementary and added a new table 1 including a summary of our estimates.

I was not able to find the value of 0.6 g-N.m².yr⁻¹ used in the manuscript in reference to Elbert et al 2012. The value found in the main manuscript and in Sup. Info. for Cryptogamic ground cover in desert ecosystems was established at 0.76g.m².yr⁻¹. Was the value of 0.6g-N.m².yr⁻¹ recalculated from a subset of the literature in the Sup. Info Table S11 of Elbert et al? Elbert et al 2012 encompasses numerous references with lichen and mosses which could indeed be used for this estimation, but it is important to explain which study has been incorporated based on specific criteria and what indicator of centrality (geometric mean, arithmetic mean, ...) has been used.

We unfortunately propagated a mistake from the Barger et al. book chapter that reported the value from Elbert et al. to be of 0.6 g-N m² yr⁻¹. This has now been corrected throughout the text, figures and tables. Because this flux value is higher than the one initially considered it even strengthens our previous conclusion. The new Figure 2 highlights that all but one estimate (Richert et al., 1978) intersects modern time export flux at low coverage percentage of Archean land surface.

REVIEWERS' COMMENTS:

Reviewer #4 (Remarks to the Author):

The manuscript in its latest form makes a strong case that continental microbial ecosystems were globally important for fixing atmospheric nitrogen and supplying fixed nitrogen to the oceans. This process, widely recognized for its role in global N cycling today but largely ignored for the Precambrian, has important implications for our understanding of the evolution of this major biogeochemical cycle from the earliest biosphere to today. There is little doubt that this work will have a significant impact on the scientific communities working on biogeochemical cycling and its evolution at diverse timescales in Earth history.

The authors have taken reviewer 3's comments seriously and made several significant changes to address them. These include addition of a Table (new Table 1) that tabulates the various estimates and their uncertainties for nitrogen outputs from biological soil crusts to soils, and an expanded discussion in this regard to make the magnitude of these uncertainties clear. The impact that these uncertainties have on the final conclusion is minor – the authors clearly demonstrate that by nearly all estimates for the transfer of nitrogen to modern soils by biological soil crusts, this process must have been important throughout Earth history, and the original conclusions stand.

The authors address a minor point of Reviewer 3 regarding abiotic sources of fixed N (e.g., generated by lightening) by citing estimated fluxes from previous work in their response letter (which prove to be much lower than the BSC fluxes in question).

I suggest only a few minor changes that are unlikely to require re-review:

New Table 1: The units of the second column are not clear – is it in Tmol / m² / yr ?

Line 241- 242: it would be useful to explicitly state that the flux of fixed N from BSCs would be significantly larger than from abiotic sources. As written, the authors simply state that the BSC flux could have helped overcome nitrogen limitation, and do not explicitly state that the BSC flux would have been much larger than the abiotic flux. Also, Stueken et al. (2016, Earth Science Reviews) provide a more complete overview of estimated abiotic fixed nitrogen fluxes (their Table 2) that would be worth citing here.

In short, I believe that this manuscript is highly mature and that the authors have sufficiently addressed the criticisms brought up in the last round of review. In my opinion this manuscript is nearly ready for publication as is, and my suggested changes are simple enough that I wouldn't expect that they would require further external review.

RESPONSES TO THE REVIEWERS COMMENTS:

We provide below a thorough answer to questions raised by the reviewer #4 (original text is provided first in black followed by our responses in green). We carefully took into account all the remarks made by the reviewer and are enclosing a revised version of our manuscript implementing these changes.

REVIEWERS' COMMENTS:

Reviewer #4 (Remarks to the Author):

The manuscript in its latest form makes a strong case that continental microbial ecosystems were globally important for fixing atmospheric nitrogen and supplying fixed nitrogen to the oceans. This process, widely recognized for its role in global N cycling today but largely ignored for the Precambrian, has important implications for our understanding of the evolution of this major biogeochemical cycle from the earliest biosphere to today. There is little doubt that this work will have a significant impact on the scientific communities working on biogeochemical cycling and its evolution at diverse timescales in Earth history.

The authors have taken reviewer 3's comments seriously and made several significant changes to address them. These include addition of a Table (new Table 1) that tabulates the various estimates and their uncertainties for nitrogen outputs from biological soil crusts to soils, and an expanded discussion in this regard to make the magnitude of these uncertainties clear. The impact that these uncertainties have on the final conclusion is minor – the authors clearly demonstrate that by nearly all estimates for the transfer of nitrogen to modern soils by biological soil crusts, this process must have been important throughout Earth history, and the original conclusions stand.

The authors address a minor point of Reviewer 3 regarding abiotic sources of fixed N (e.g., generated by lightening) by citing estimated fluxes from previous work in their response letter (which prove to be much lower than the BSC fluxes in question).

I suggest only a few minor changes that are unlikely to require re-review:

New Table 1: The units of the second column are not clear – is it in Tmol / m² / yr ?

It is a percentage: “Percentage of Archean land coverage needed to reach the modern N export flux”. We removed the 0.15 Tmol / yr value appearing in the heading of column two which is referring to the modern export flux of N (this value is provided in the caption of the figure) and could introduce confusion with the percentage metric used in this column.

Line 241- 242: it would be useful to explicitly state that the flux of fixed N from BSCs would be significantly larger than from abiotic sources. As written, the authors simply state that the BSC flux could have helped overcome nitrogen limitation, and do not explicitly state that the BSC flux would have been much larger than the abiotic flux. Also, Stueken et al. (2016, Earth Science Reviews) provide a more complete overview of estimated abiotic fixed nitrogen fluxes (their Table 2) that would be worth citing here.

Modified accordingly: Line 294 “However, the flux of fixed N from BSCs would be significantly larger than that from abiotic sources⁷⁰ and it could have helped overcome the limitation of early biosphere to fixed N using diazotrophy⁷¹ and fuel primary productivity with newly available nitrogen species.”

In short, I believe that this manuscript is highly mature and that the authors have sufficiently addressed the criticisms brought up in the last round of review. In my opinion this manuscript is nearly ready for publication as is, and my suggested changes are simple enough that I wouldn't expect that they would require further external review.